# First laboratory confirmation and sequencing of *Zaire ebolavirus* in Uganda following two independent introductions of cases from the 10th Ebola Outbreak in the Democratic Republic of the Congo, June 2019

Luke Nyakarahuka[1,2]*, Sophia Mulei[1], Shannon Whitmer[3], Kyondo Jackson[1], Alex Tumusiime[1], Amy Schuh[3], Jimmy Baluku[1], Allison Joyce[3], Felix Ocom[4], Jayne B. Tusiime[5], Joel M. Montgomery[3], Stephen Balinandi[1], Julius J. Lutwama[1], John D. Klena[3‡], Trevor R. Shoemaker[3‡], on behalf of 'Kasese EVD Outbreak Response Team'[¶]

**1** Department of Arbovirology, Emerging and Re-emerging Infectious Diseases, Uganda Virus Research Institute, Entebbe, Uganda, **2** Department of Biosecurity, Ecosystems and Veterinary Public Health, Makerere University, Kampala, Uganda, **3** Viral Special Pathogens Branch, Division of High-Consequence Pathogens and Pathology, United States Centers for Disease Control and Prevention, Atlanta, Georgia, United States of America, **4** Uganda Public Health Emergency Operations Center, Kampala, Uganda, **5** World Health Organization, Kampala, Uganda

‡ These authors are joint senior authors on this work.
¶ Membership of the Kasese EVD Outbreak Response Team is provided in the acknowledgements.
* nyakarahuka@gmail.com

## Abstract

Uganda established a domestic Viral Hemorrhagic Fever (VHF) testing capacity in 2010 in response to the increasing occurrence of filovirus outbreaks. In July 2018, the neighboring Democratic Republic of Congo (DRC) experienced its 10th Ebola Virus Disease (EVD) outbreak and for the duration of the outbreak, the Ugandan Ministry of Health (MOH) initiated a national EVD preparedness stance. Almost one year later, on 10th June 2019, three family members who had contracted EVD in the DRC crossed into Uganda to seek medical treatment.

Samples were collected from all the suspected cases using internationally established biosafety protocols and submitted for VHF diagnostic testing at Uganda Virus Research Institute. All samples were initially tested by RT-PCR for ebolaviruses, marburgviruses, Rift Valley fever (RVF) virus and Crimean-Congo hemorrhagic fever (CCHF) virus. Four people were identified as being positive for *Zaire ebolavirus*, marking the first report of *Zaire ebolavirus* in Uganda. In-country Next Generation Sequencing (NGS) and phylogenetic analysis was performed for the first time in Uganda, confirming the outbreak as imported from DRC at two different time point from different clades. This rapid response by the MoH, UVRI and partners led to the control of the outbreak and prevention of secondary virus transmission.

**Data Availability Statement:** All relevant data are within the manuscript.

**Funding:** This study was funded by United States Centres for Disease Control and Prevention (CDC), Ministry of Health of Uganda and World Health Organization (WHO). The funders had no role in study design, data collection and analysis, decision to publish, or preparation of the manuscript.

**Competing interests:** The authors have declared that no competing interests exist.

## Author summary

In the effort to control the on-going COVID-19 pandemic, countries instituted lock downs and closed international borders. But is this the best approach to controlling trans-boundary infectious diseases? In this publication we demonstrate how we managed to control Ebola Virus Disease (EVD) introduced into Uganda from Democratic Republic of Congo (DRC) in 2019. Once the EVD outbreak was announced in DRC, we intensified cross-border surveillance and instituted acceptable public health control measures following international health regulations to limit onward community and nosocomial transmission in Uganda. Consequently, on July 10th, 2019, three (3) cases crossed into Uganda and were detected at the point of first contact with health facility in Uganda in Kasese district and the fourth case was detected in August 2019 at a point of entry at the Uganda-DRC border by temperature screening. Sequencing of these cases showed that they were independent introductions related to two different clades of the ongoing outbreak in DRC. All the patients died without onward secondary transmissions in Uganda. We invite you to read this publication and learn how this was achieved as it can be used as a model for cross border surveillance to control similar infectious disease outbreaks.

## Introduction

Four species of Ebolavirus are known to cause Ebola Disease (EBOD) and have been associated with outbreaks in humans mainly in Sub-Sahara Africa. These species include *Zaire ebolavirus*, *Sudan ebolavirus*, *Bundibugyo ebolavirus* and *Taï Forest ebolavirus*. Other Ebolavirus species have been identified in wildlife but until now, have not yet been associated with disease among humans [1]. Two of these species are *Reston ebolavirus* and *Bombali ebolavirus*, the latter of which includes a recently described filovirus from Sierra Leone. Uganda first reported an EBOD outbreak in 2000 caused by Sudan virus; followed by another outbreak in 2007, caused by a novel ebolavirus, Bundibugyo virus, in the ranges of Mount Rwenzori in Bundibugyo district [2–4]. Three additional Sudan Virus Disease (SVD) outbreaks were reported in 2011 (Luweero) and in 2012 (Kibale and Luweero districts) all caused by Sudan Virus [5]. Uganda had not reported an outbreak of Zaire Ebola virus, which primarily occurs in Central and Western Africa and is considered the most severe forms of all the Ebolavirus infections due to its high case fatality rate (CFR) [6].

Uganda is at high risk of filovirus outbreaks according to risk mapping studies using environmental conditions that favor the survival of filovirus reservoirs [7]. For this reason, the country has developed the capacity to detect and respond to pathogens of high consequence through the establishment of a VHF surveillance program, a collaboration between Uganda Virus Research Institute (UVRI) and the Viral Special Pathogens Branch of US Centers for Disease Control and Prevention (CDC). As part of this program, a VHF diagnostic laboratory was established together with a supportive sentinel surveillance system for the rapid detection of VHFs and to initiate early response [8]. One of the first established VHF sentinel surveillance sites, Kagando hospital, is in Kasese district neighboring the Democratic Republic of Congo (DRC). Kagando hospital was chosen because of its strategic location in Western Uganda, the location of several previous outbreaks of VHF, plus its proximity to DRC, where introduction events from VHF outbreaks in DRC would likely occur due to human population movement. At this site, UVRI pre-positioned VHF surveillance supplies and trained personnel to rapidly identify any VHF suspect case and safely collect samples and transport them to the Uganda National VHF Reference Laboratory at UVRI in Entebbe.

On the morning of 10[th] June 2019, a suspect case of EVD who had just crossed the DRC-Uganda border was identified by triage medical staff 30 kilometers away at Kagando hospital. A sample was collected by trained personnel using the prepositioned personal protective equipment (PPE) and supplies and transported on the same day to UVRI for molecular diagnostic testing. The following morning, this sample was confirmed as positive for *Zaire ebolavirus* by RT-PCR. UVRI subsequently notified the Ministry of Health (MOH) through the Public Health Emergency Operations Centre (PHEOC). Immediately following this notification, MOH officially declared Uganda's 6[th] outbreak of EVD on the 11[th] June 2019. This was the first time Uganda detected and reported an acute case of *Zaire ebolavirus*. A multidisciplinary rapid response was initiated upon this confirmation to control the outbreak. On 12[th] June 2019, two additional cases of *Zaire ebolavirus* were confirmed, all belonging to the same family which had crossed into Uganda from Beni in DRC. On 29[th] August 2019, a fourth EVD case, also epidemiologically linked to the 10[th] DRC EVD outbreak, was confirmed by the UVRI VHF laboratory. This case was not related to the three previously detected cases in June following genomic sequencing of the confirmed case samples. We describe here the details of the laboratory testing and epidemiological findings that led to the rapid identification and containment of this outbreak.

## Material and methods

### Ethics statement

This was an outbreak investigation and public health emergency that was approved by the National Task Force (NTF) for disease outbreaks in Uganda, and hence it was considered a non-research activity and waived from institutional review board or ethics committee. All persons who were interviewed and whose samples were tested gave informed oral consent, and a parent or guardian of any child participant provided informed consent on the child's behalf.

### EVD case detection

Prior to the 10[th] EVD outbreak in DRC, a suspect VHF case definitions was distributed, and health workers were trained on how to identify VHF suspects at the established VHF sentinel surveillance sites and in all other health facilities in Uganda. The suspect VHF case definition was: any patient with acute illness presenting to a health facility with fever or history of fever and no alternative diagnosis (e.g. malaria, typhoid fever, brucellosis), with at least four of the following signs/symptoms: vomiting, diarrhea, muscle or joint pain, chills/rigors, intense fatigue, abdominal pain, skin rash, difficulty in swallowing, headache, unexpected bleeding from any site, jaundice (yellowing of eyes or any other mucus membranes). Because there was an ongoing EVD outbreak in North Kivu province in neighboring DRC, any person with fever and any of the above symptoms plus an epidemiological link to a confirmed or suspect case in DRC was tested for EVD regardless of alternative diagnosis. Samples were also submitted from other districts in Uganda that were not bordering DRC during this period.

A confirmed case was one that tested positive by RT-qPCR for Ebola virus. Epidemiological data was collected on the standardized VHF case investigation form by patient interview and a blood sample collected following biosafety and biosecurity measures by trained laboratory staff at Kagando hospital for the first suspect case. Three more EVD cases were detected at Bwera hospital, and samples were collected, triple-packaged, and sent to the UVRI VHF reference laboratory for confirmation. After declaration of the EVD outbreak, investigations were performed on all confirmed cases. Contacts of confirmed cases were line-listed using contact tracing forms and data were managed in the field using the EpiInfo VHF application [9] and the newly developed *Go.Data* software [10]. Contacts were followed daily for 21 days and ring

vaccination initiated for contacts and contacts-of-contacts as previously recommended for curtailing possible chains of transmissions [11].

## Laboratory testing

**Polymerase chain reaction, IgM and antigen detection.**   Three to five milliliters of blood were obtained from all suspected cases of a VHF for laboratory testing using either RT-qPCR or ELISA at the UVRI VHF laboratory according to established protocols [12–14]. All samples submitted from all parts of the country including those from high-risk areas were subjected to PCR as a gold standard for testing VHFs. All samples submitted were tested for diagnosis of six viral species that cause VHFs which include Sudan Virus, Ebola virus, Bundibugyo virus, Marburgviruses, Crimean-Congo Hemorrhagic Fever virus and Rift Valley Fever Virus. This is how other outbreaks of VHFs were detected alongside surveillance for EVD.

Briefly the PCR procedure is as follows; RNA was extracted from whole blood using 5X Magmax 96 Viral Isolation kit (Applied Biosystems Inc., Vilnius, Lithuania) according to manufacturer's instructions. Subsequent RT-PCR assays targeted the Ebola virus NP viral gene. ELISA for Ebola virus antigen and anti-Ebola virus IgM detection was performed using 96-well plates. Unless otherwise stated, all ELISA procedures used 100μl test and reagent volumes per well format; plates were washed 3 times using 0.1% Tween-20 in phosphate-buffered saline (PBS)-v/v between all procedures; and all incubation temperatures were at 37˚C for 1hour. In addition, all reagents used in all procedures were diluted in PBS containing 5% skim milk.

For serological testing, we used in-house ELISA assays developed and used at US CDC for over 20 years. The details of the method can be found in MacNeil *et al* (2011)[15]. Additional references to the development of the method can be found in Ksiazek *et al* (1999) [12,13] and Onyango *et al* (2007) [14].

Briefly, both the IgM and IgG ELISA assays utilize inactivated, sonicated Ebola infected Vero E6 cells (positive-antigen) as the source of Ebola antigen; inactivated, sonicated non-infected Vero E6 cells (negative-antigen) are used to control for the Vero E6 background. Each patient serum sample is tested at four 4-fold dilutions, covering the range from 1:100, 1:400, 1:1600 and 1:6400. At each titration point the negative-antigen reading is subtracted from the positive-antigen reading to provide an adjusted optical density (OD410nm) reading for that titration point. Adding together all the adjusted OD410nm readings for each titration provides a sum OD410nm value. For a sample to be considered IgM positive, it must have a minimum titre of 1:400 (an IgM titre is considered positive if the adjusted OD410nm is >0.1) and a minimum sum OD410nm value of >0.45. For a sample to be considered IgG positive, it must have a minimum titre of 1:400 (an IgG titre is considered positive if the adjusted OD410nm is >0.2) and a sum OD410nm value of >0.95.

**Next generation sequencing.**   Ebola virus (EBOV)-positive specimens were prepared for sequencing at the UVRI VHF laboratory and additional virus isolation and sequencing at the Viral Special Pathogens Laboratory (CDC), Atlanta, GA, USA, following a protocol developed by CDC. Briefly, whole blood specimens were inactivated with Tripure (Roche, Basel Switzerland) or 5X Magmax 96 Viral Isolation kit (Applied Biosystems Inc., Vilnius, Lithuania). Tripure-extracted RNA was phase separated with 1-bromo-3-chloropropane and applied to Clean and Concentrate-25 (Zymo Research, CA, USA) columns for further purification and concentration. RNA was treated with RNase-free DNase (Roche) and prepared for unbiased next generation sequencing (NGS) using the NEBNext Ultra II Directional RNA library preparation kit (NEB). PhiX was added to pooled libraries (1%) and libraries were sequenced using an Illumina iSeq (V1 2 × 150 cycles) or MiSeq (High Output 2 × 150 cycles). Reads were mapped to

an EBOV Ituri reference genome (MK007329) using in-house scripts—consisting of adaptor removal (cutadapt), quality trimming (printseq-lite -min_qual_mean 25 -trim_qual_right 20 -min_len 50), read mapping (BWA-mem), PCR-de-duplication (picard MarkDuplicates) and consensus genomes were called using Geneious (threshold = 0%, Assign Quality = total, minimum coverage > 2; version 10). Evolutionary history was inferred with EBOV genomes from Genbank and INRB [16–18] ((using raxml (-m GTRGAMMA -p $RANDOM -f a -x $RANDOM -N 1000) with bootstrap support provided by 1000 iterations. EBOV genomes were deposited to Genbank: MZ854250-3.

## Results

### Epidemiological investigations

Since the declaration of the 10[th] EVD outbreak in DRC (July 2018) until December 2019, 1098 VHF suspect samples were tested at the UVRI VHF reference laboratory of which 530 (48.3%) were from the 'high-risk' districts bordering the DRC, as mapped by MoH and other partners (Figs 1 and 2). All of these samples had tested negative for Ebola virus until 10 June 2019 when a 5-year-old male (CASE 1) was confirmed as infected with EBOV. Two additional related cases (CASE 2 and CASE 3) were confirmed as EBOV on 12 June 2019; all from the same family and having crossed the border into Uganda from DRC at the same time. Two months later, another spillover case of EBOV from DRC was confirmed on 29[th] August 2019 (CASE 4), totaling to 4 EVD cases during this enhanced surveillance period. All the cases were detected from Kasese district that borders DRC, having one of the major points of entry (POE).

The VHF surveillance system did not only detect the four acute EVD spill-over cases, but additional acute VHF cases. There were a total of 18 CCHF and 10 RVF cases that were laboratory confirmed during the surveillance period, between 1[st] August 2018-December 2019. On average, a case of a VHF was confirmed by the UVRI VHF laboratory every month, but case detection peaked around the months of August 2018 and June 2019 (Fig 3)

Fig 4 summarizes the timeline of events that led to the first detection of EBOV in Uganda. The events began with the death of the grandfather (BN-001) of CASE1 in Beni territory in

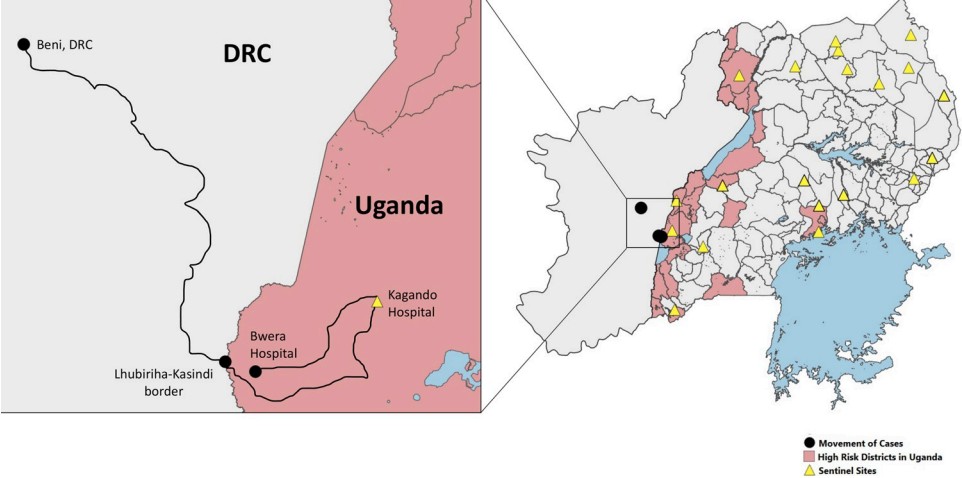

**Fig 1. A map of Uganda showing the movement of cases, high-risk districts, and location of VHF sentinel surveillance sites.** The black dots connected by lines show the movement of cases, districts designated as high-risk for EBOV spillover are colored pink, and VHF sentinel surveillance sites are indicated by yellow triangles (Shape file Source: www.diva-gis.org/gdata and https://energydata.info/dataset/africa-water-bodies-2015).

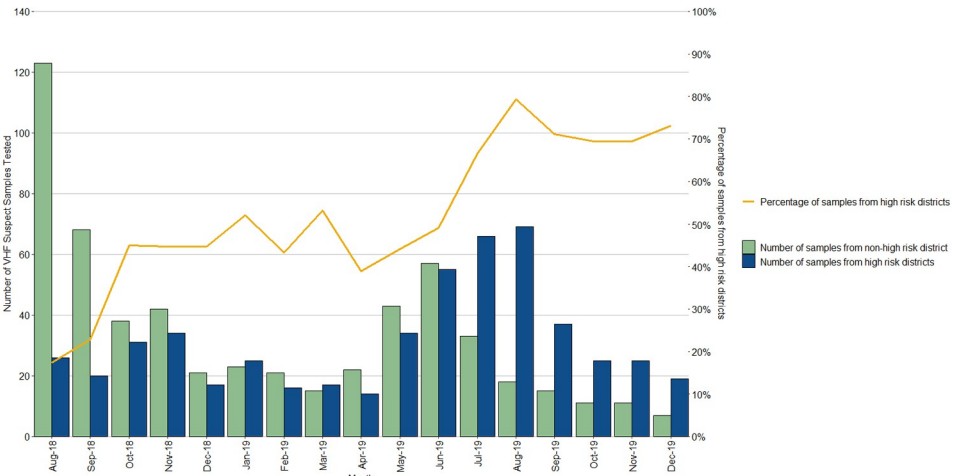

**Fig 2. Viral hemorrhagic fever suspect samples tested since the commencement of EVD preparedness activities in Uganda 1st August 2018 –December 2019.**

DRC. The daughter of BN-001, a 28-year-old mother (not a case) of three children which included the 5-year-old male (CASE1), 3-year-old (CASE2) and 1.5-year-old (not a case) was married to a Ugandan and the family previously lived in Kasese district, Kirembo village. The parents had not lived together for 1.5 years with the mother along with the three children living with BN-001 family in Beni territory across the border in DRC, while the father remained in Uganda. BN-001 who was a pastor by occupation, reported symptom onset on 29th May 2019 in DRC. BN-001 was nursed by the wife (CASE3) and other family members including CASE1 and CASE2 who were living together in Beni territory DRC. BN-001 had an additional two homes in DRC, Beni territory, in Masambo village and another Lhubiriha-Kasindi border town near Uganda.

On 1st June 2019, BN-001's clinical condition rapidly deteriorated and he died on the same day, alerting the DRC EVD surveillance teams to suspect EBOV infection. A swab was

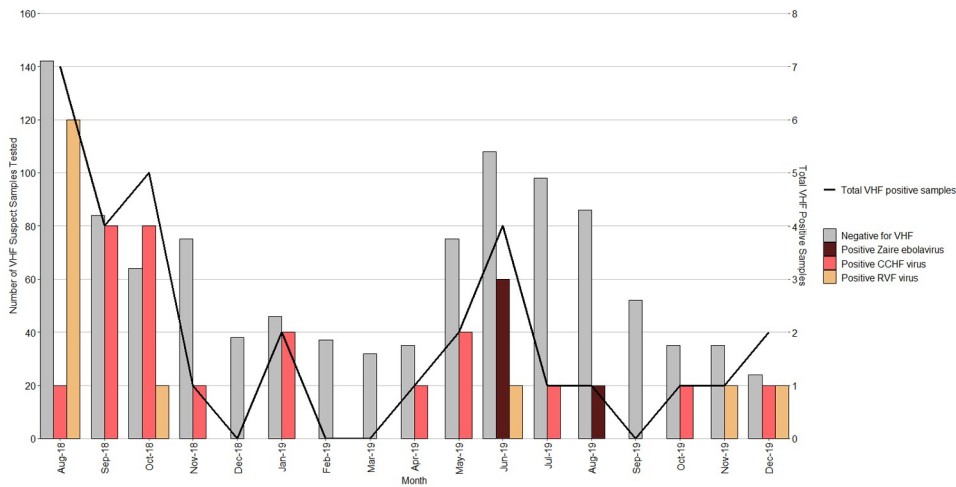

**Fig 3. VHF pathogens detected during first year of EVD preparedness period in Uganda 1st August 2018 – 30th December 2019.**

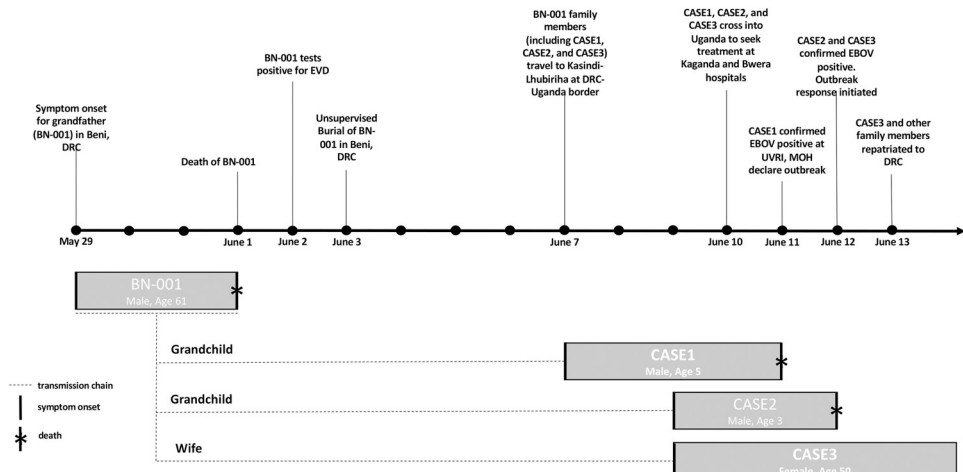

**Fig 4. Timeline of events leading to first detection of *Zaire ebolavirus* in Uganda, June 2019 and the transmission chain.**

collected and tested on 2$^{nd}$ June 2019 by the DRC EVD investigation team and was found positive for EBOV. BN-001 was buried on 3$^{rd}$ June 2019 in an unsupervised burial in Beni-DRC.

On 4$^{th}$ June 2019, a group of 11 people from Kasese district who are relatives of the affected family in DRC travelled to Masambo village in DRC for funeral rites ceremony but did not attend the actual burial of BN-001, which had occurred in Beni the previous day, 3$^{rd}$ June 2019. They did not come in contact with the body of BN-001 nor his contacts as they were still in Beni. Most returned on the same date 4$^{th}$ June 2019, yet a few returned on 6$^{th}$ June 2019. All family members that travelled from Uganda to DRC for funeral rites reported not having contact with either BN-001 or his contacts and none of them contracted EVD.

On 5$^{th}$ June 2019, the mother to CASE1 and CASE2, along with other family members including CASE3, travelled from Beni in DRC via their ancestral home in Masambo. On 7$^{th}$ June, they travelled to Lhubiriha-Kasindi, a border town at the border with Uganda.

On 8$^{th}$ June 2018, CASE1, CASE2 and their grandmother—CASE3 became gravely ill while in border town of Lhubiriha-Kasindi, DRC. The mother of CASE1 and CASE2, having been married to a Ugandan, who was also the father of CASE1 and CASE2 contacted her ex-husband to organize transfer of the sick children to Uganda for treatment and care.

On 10$^{th}$ June 2019, the mother together with CASE1, CASE2 and CASE3 all crossed into Uganda. The mother carrying sick CASE1 on her back hired a motorcycle taxi commonly used for transport and locally known as a '*bodaboda (*BB*)*' to transfer them and cross the border into Uganda to Kagando hospital via Kirembo village, Kisinga sub county, Kasese district, leaving CASE2 and CASE3 at Kasindi village on the DRC side. The mother of the children hired BB1 in 'no-man's-land' on the DRC side and they crossed into Uganda via the official point of entry. She reported that she informed the security officials guarding the entrance into Uganda that she was just transferring the child from across the border to the hospital and she was allowed to pass since the EVD screening team had not yet arrived at 7.00am. After crossing the border into Uganda, she was driven with the baby on her back by BB1 to the home of the father to CASE1 in Kirembo village in Uganda, near Kagando hospital where they had a stop over to collect money for hospital fees and other utilities for hospital admission. At this home, there were only two contacts, one aunt who picked a wet piece of cloth that was being used to cool down the temperature of CASE1 and the paternal grandmother who gave them money for hospital fares. They reportedly spent approximately less than an hour at this home as they

were rushing to take the baby to the hospital. Upon arrival at Kagando hospital, they were joined by two other relatives: an uncle who carried the baby from the motorcycle to hospital outpatient department (OPD) and an aunt that helped them at admission. In the OPD at Kagando hospital, they were received by one male nurse who carried CASE1 to an attending clinician, making two contacts among the hospital staff, of which one had already been vaccinated against EVD. Another family member had notified the hospital of how CASE1 could be an EVD suspect and immediately the clinicians isolated CASE1 before admission on the general ward. A sample was safely collected from CASE1 by the Kagando laboratory hospital staff using already prepositioned sample collection supplies and PPE provided by UVRI VHF surveillance program. A sample was sent to UVRI for testing on the same day. At 12:30pm the same day, Bwera hospital Ebola Treatment Unit (ETU) ambulance was called to transfer CASE1 and his mother to the EVD ETU at Bwera hospital.

At around 2pm, the mother was interviewed by health workers at Bwera ETU and she was asked to bring in other ill family members across the border for treatment, which was granted. She hired another *bodaboda* (BB2) stationed outside Bwera hospital and it transported her across the border back to Kasindi village, DRC to pick up CASE2 and CASE3. She hired two other *bodabodas* (BB3 and BB4). BB3 carried the mother and CASE2 from Kasindi-Lhubiriha straight to the ETU at Bwera whereas BB4 carried CASE3 also straight to Bwera hospital. However, BB4 took CASE3 first to the OPD where she met her sister who was waiting late on the evening of 10th June 2019. CASE3 remained on the general ward at Bwera hospital until she was identified by Bwera hospital staff and transferred to the ETU on the following day of 11th June 2019 at 2pm. A sample was collected from CASE2 and CASE3 on 11th June 2019 and sent to UVRI for testing. CASE1 died on 11th June 2019 with severe bleeding symptoms among other VHF clinical symptoms. On the morning of 12th June 2019, CASE2 and CASE3 were confirmed positive for EBOV by the UVRI VHF laboratory. CASE3 died on the evening of 12th June 2019 without exhibiting any bleeding symptoms. On the morning 13th June 2019, the mother together with confirmed CASE2 were repatriated back to DRC together with other family members. A total of 114 contacts were listed and followed for 21 days with none of the contacts becoming secondary cases. However, all consenting and qualified contacts, contacts of contacts and frontline health workers were subsequently vaccinated using the rVSV-ZE-BOV EBOV vaccine [11]. Following confirmation of these cases, an emergency National Task Force (NTF) was called in the evening of 11th June 2019 and National Rapid Response Teams (NRRT) were constituted and an EVD outbreak was officially declared.

Fig 4 shows the family relationships of spillover cases in June 2019 into Uganda from DRC. The source case (BN-001) was a resident of Beni DRC and he was head of the family who first fell sick. It is believed that he contracted the infection from praying and touching parishioners, since he was a pastor by profession. He was cared for by his wife (CASE3) and his grandchildren (CASE1 and CASE2) who were detected in Uganda as they came to seek medical treatment. CASE4, who was detected later in August 2019 by temperature monitoring at the Mpondwe point of entry (POE), was not related to the first importations (CASE 1–3). This is further explained in Fig 5 where the sequence of CASE4 (201901815- MZ854253) detected in August, which is not related to first cases (CASE1-201901276, CASE2-201901277 and CASE3-201901278) detected in June 2019 but belonged to the Katwa clade of virus that was circulating in that region, that originated from Beni, DRC. All four confirmed cases in Uganda were closely related to the strains of EBOV that were circulating in DRC at that time confirming this as an imported outbreak into Uganda from DRC. All four cases entered Uganda and subsequently died, giving a case fatality rate 100%. Serological testing revealed that none of the cases had mounted an immune response as no antibodies (IgM or IgG) against EBOV were detected in blood samples.

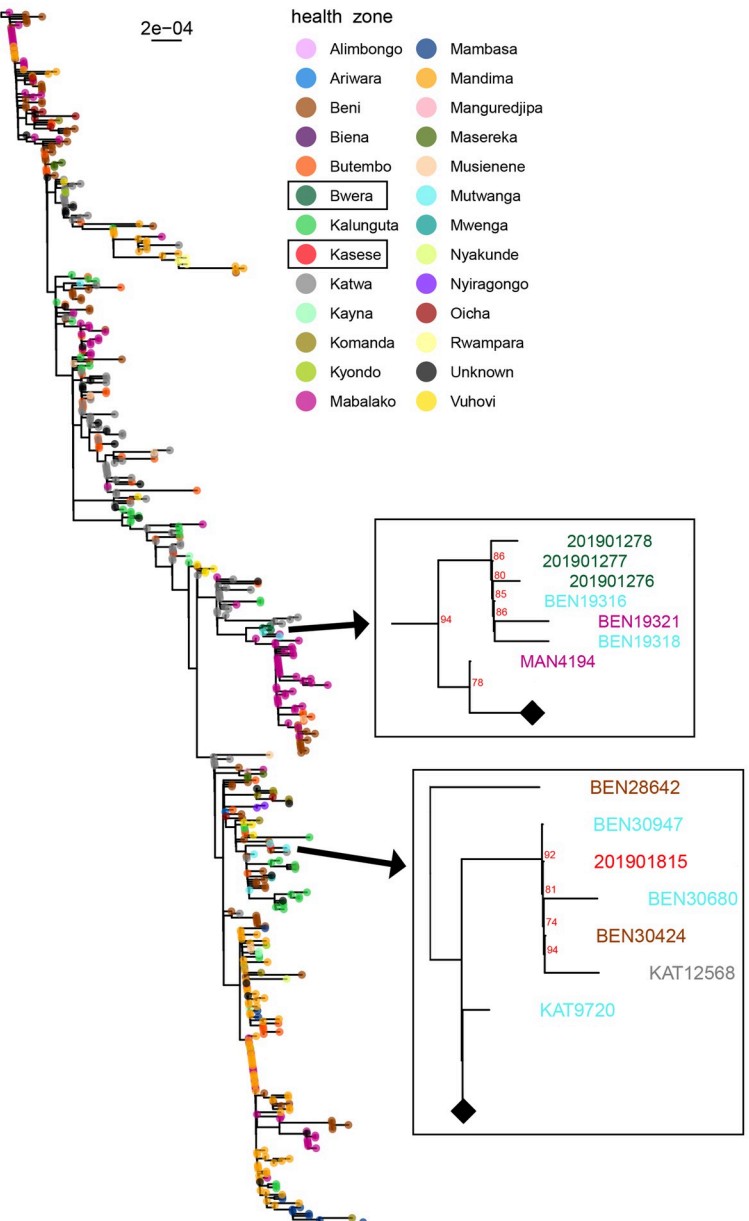

**Fig 5. Phylogenetic tree showing relationship between the imported cases in Kasese district, Uganda with Source Transmission chains in DRC: Specimen number (Genbank accession number): 201901276 (MZ854250), 201901277 (MZ854251), 201901278 (MZ854252), 201901815 (MZ854253).**

## Discussion

The DRC declared an outbreak of EVD on 31st July 2018 in the territory Beni, 30km away from the Uganda border. A risk assessment showed that Uganda was at high risk for imported cases, especially via border crossing into Uganda either for business, health care, migration or family visitation. An EVD preparedness plan was put in place and the VHF reference laboratory at UVRI was an integral part of this plan [19]. Suspected VHF samples were collected from all over the country, including those from districts categorized as high-risk, and sent to the UVRI VHF laboratory for testing. From August 2018 to December 2019, 1093 VHF

suspect samples were submitted of which 48% (530/1093) were from high risk districts, 8.4% (92/1093) were from VHF sentinel surveillance sites and 2.9% (32/1093) tested positive for at least one VHF pathogen. This preparedness activity began at the peak of increased RVF transmission within 52 districts located within the cattle corridor of Uganda, hence an amplification for high numbers of suspect samples submitted in the laboratory within the month of August 2018 (Fig 1). The proportion of samples tested from high-risk districts was low (17%) at the beginning of the EVD preparedness activities in August 2018, but later increased steadily to 67% in July 2019 involving the detection and confirmation of EBOV in June 2019, almost one year since declaration of the EVD outbreak in neighboring DRC. This increase in samples from high-risk districts in Uganda corresponded with the increase in the number of confirmed EVD cases in the DRC at that time. An increase in cases in DRC, especially near the Ugandan border, greatly increased the risk of having potential introduction of cases and hence the need to increase preparedness and investment in the surveillance and diagnostic capabilities in these areas. Fig 2 shows that most of the samples tested at UVRI were not from high-risk districts at the beginning of the preparedness activity, only two months between January and March 2019 had higher number of VHF suspect samples submitted from high-risk districts. This allowed the VHF laboratory to confirm other non-filovirus VHF outbreaks, while focusing testing on EVD suspect samples. These included 16 independent human cases of CCHF and 8 human cases of RVF from districts not categorized as high-risk (Fig 2). The number of suspect VHF samples drastically increased, especially from high-risk districts, when a case of EVD was confirmed on 11th June 2019. This is common with EVD outbreaks as the index of suspicion tends to increase with detection of any confirmed cases. However, an increase in samples submitted for VHF testing is also related to the broad case definition of VHF suspects, as they tend to present like any other common infectious disease seen in the tropics such as malaria, typhoid and other acute febrile illness. Most of the samples initially suspected as VHF patients were negative, between June and July 2019 when 332 samples were submitted, and only five of these samples tested were positive for a VHF (3 EVD, 1 CCHF and 1 RVF). For this reason, there have been efforts to describe alternative diagnoses for these non-VHF suspect samples, but preliminary findings from testing on Taqman array card technology on human pathogens that circulate in East Africa show that most of these samples were positive for malaria parasites as opposed to VHFs [20]. Hence there is need to discuss and improve the case definition for VHF suspects or design cheap, rapid, and high-throughput diagnostic approaches that can rule-out infection with commonly circulating pathogens prior to testing for VHFs. Testing high numbers of samples under high containment laboratory conditions is both expensive and labor intensive in terms of biosafety and biosecurity concerns.

In Fig 2, we observe that filoviruses are not the only pathogens of concern as the etiologic agents of VHFs. Samples submitted and thought to be highly suspect of EVD or Marburg Virus Disease have most of the time turned out to be RVF or CCHF infections. Alongside heightened surveillance, we have confirmed 24 non-filovirus VHFs, including 16 CCHF and 8 RVF viral infections in humans (between August 2018—December 2019). This shows the importance of a broad-based testing algorithm at UVRI to identify and test for an array of etiologic agents of VHF at the same time. Out of 12 months of EVD preparedness, we have detected at least one VHF per month, apart from the months of December 2018, February and March 2019. Any VHF case confirmed has a positive feedback mechanism in terms of samples submitted because the index of suspicion tends to increase when a confirmed case is detected. Positive VHF cases other than filoviruses provide us with an opportunity to train and prepare for more infectious agents such as EVD, simply because the infection prevention and control requirements are the same. It also shows that health workers are aware of the case definition of a VHF.

Detection of three EVD cases in June 2019 was facilitated primarily by two main points: One was the cross-border collaboration and communication with the DRC team that had sent the contact list containing all the three confirmed cases to the Uganda surveillance teams. These cases had been isolated in DRC near the Ugandan border as they were contacts of a known confirmed case. However, according to the patients' judgment, they believed they would get better care if they crossed to into Uganda. The second point is that the patients first presented at a VHF sentinel surveillance site at Kagando hospital that was able to quickly identify and respond to them as soon as they arrived, limiting the exposure of healthcare workers to only two triage staff. This was crucial to limit any secondary transmission in the hospital setting that would have complicated the contact tracing and magnitude of this importation. Kagando hospital had been well prepared for such a scenario since 2011 when it was established as one of the first UVRI VHF sentinel surveillance sites with a trained clinical and laboratory team who act as mentors for other health workers to detect suspect VHF patients. Once the suspect patient was detected, he was taken to isolation at Kagando hospital together with his mother and a blood sample was collected safely without any further high-risk contact with the patient. Immediately the Kasese district EVD team was notified by the Kagando hospital team and the suspect patient was transferred to Bwera hospital ETU, which was prepared to handle any EVD cases in the region in case of spill-over from DRC. This was a very good example of a coordinated effort to swiftly reduce transmission in a hospital setting and none of the two health workers that came into contact with the patient at Kagando hospital became a case, neither were any of the contacts listed during this investigation. Whether this is attributed to the EVD vaccination effort of front-line healthcare workers cannot be fully ascertained, as more studies are still being conducted on the effectiveness of the vaccine. The VHF preparedness efforts through a sentinel surveillance program helped greatly in making sure the number of high-risk contacts was limited which often occur if patients are nursed for a long period at the hospital or by family members and can be exacerbated if unsafe traditional burial practices are also performed.

Another factor that contributed to the success of this outbreak detection and response was the presence of the pre-existing VHF diagnostic capacity in Uganda. For example, the first confirmed case in this outbreak was detected with a laboratory turn-around time (TAT) of 4 hours. This excludes the transportation of samples from Kagando hospital to Entebbe UVRI VHF laboratory which, if included was less than 10 hours. However, the TAT for alert and suspect cases of VHF was further shortened after the immediate deployment of the mobile field diagnostic laboratory in Kasese district upon confirmation of the first cases [21]. All samples were tested for EBOV by GeneXpert at the mobile field diagnostic laboratory, and a duplicate sample was sent to VHF Program at UVRI for PCR testing against a panel of VHFs. Samples confirmed to be positive for EBOV were tested by ELISA for viral antigen and antibody and sequenced using followed Illumina next generation sequencing. This was the first time that sequencing of ebolavirus samples were performed concurrently during an acute outbreak investigation in Uganda, further indicating the capacity developed by the UVRI VHF surveillance and laboratory program in preparedness and containment of filovirus and other emerging epidemics.

Sequencing results as indicated in Fig 5, showed a close relationship between EBOV clades seen in Uganda with those of Mutwanga/Mabalako (Case 1–3, 20191276–8) and Mutwanga/Beni/Katwa (Case 4, 201901815) DRC. Katwa is a town only 40 km from the Uganda border with DRC, confirming that the Ugandan outbreak was imported from DRC. This is supported by the fact that Uganda has a VHF surveillance system that had not previously detected EBOV circulating in Uganda. Also, Uganda has reported more outbreaks of EVD only second to DRC, but these had been attributed to Sudan and Bundibugyo strains. Whereas the ecosystem

of the current outbreak in DRC is similar to that on the Ugandan side of the border, the distribution of different filovirus species is likely determined by the distribution of the reservoir, the search for which is still ongoing. Also to note is that DRC has never reported an outbreak of Sudan virus, however, both Uganda and DRC have reported Bundibugyo virus [4,5]. From Fig 5, we notice that the case detected later in August 2019 (201901815) was from a different lineage from the three related cases in June 2019, demonstrating that the two Uganda EVD outbreaks were independent introductions and not the result of ongoing EBOV transmission in Uganda.

Control of filovirus outbreaks is based on detecting them early enough so that control interventions can be put into place prior to community or hospital transmissions. Cross-border collaboration is very important to control epidemics since infectious diseases do not respect international borders. Fig 3 demonstrates that events that led to the outbreak in Uganda originated in DRC. The DRC team had informed the Ugandan team of the high-risk contacts that may cross into Uganda. As soon as the cases crossed into Uganda, they were quickly detected. The fourth EVD case that occurred on 29th August 2019 was detected by Kasese PoE screening staff, showing the importance of effective EVD screening at PoEs. This was because this case crossed the PoE during working hours and his elevated temperature was detected by the thermal-scanner and he was subsequently taken by ambulance directly to Bwera ETU limiting further chains of transmission. There is a need to establish 24-hour screening for EVD suspects, but this may not be adequate to detect all suspect cases since people may be incubating the virus and can be missed by the temperature monitoring performed at the border.

Fig 4 show that cross border surveillance can be an important part of stopping the transmission of infectious disease such as EVD. We see that events in Uganda happened as a result of uncontrolled transmission in the neighboring DRC, Beni territory. Although the outbreak in DRC was still ongoing, the DRC team informed Uganda of possible contacts that could have crossed the border. Ugandan officials were ready to detect any cases that crossed, and these cases were detected as soon as they crossed. Neighboring countries could significantly reduce transborder infections by improving 'border health' collaboration. In-country transmission would have been very high if Uganda was not aware of the contacts that had crossed over, highlighting the importance and need for preparedness.

In conclusion, the Uganda VHF surveillance program and the UVRI VHF laboratory was able to quickly detect and confirm the first cases of EBOV in Uganda as soon as the cases entered the country. Following this, a mobile laboratory was rapidly deployed to the outbreak area. In addition, the VHF laboratory was able to conduct simultaneous NGS to support the outbreak response and case investigations. Working with international and other non-governmental partners, Uganda was able to prevent secondary transmission chains and limited this EBOV importation to only four cases in 2019, all of which were initially infected in DRC but traveled to Uganda to seek treatment. A robust Uganda surveillance program for VHFs is critically important and serves as an example to other countries faced with the threat of continued VHF outbreaks.

## Acknowledgments

### Kasese EVD outbreak response team

These include Atek Kagirita⸴ Yusuf Baseke, Loyce Kabyanga, Samuel Muhindo, Costantino Thembo, Columbus Masereka, Emir Talundzic, Markus H Kainulainen, Ketan Patel, James Graziano, Raymond Mugabe, Amy Whitesell, James Fuller, Elizabeth Ervin, Caitlin M. Cossaboom, John Kayiwa, Aliddeki Maria Dativa, Leocadia Kwagonza, Mutegeki Kahuka, Ben

Masiira, Collins Mwesigye, Mutoro Julius, Mbambu Agnes, Mulere Nelson, Badaki Morris, Muhesi Abraham, Mirembe Bernadette Basuta, Bernard Lubwama, Issa Makumbi, Milton Wetaka, Joseph Ojwang, Allan Muruta, Henry Mwebesaʾ Vance Brown, Jaco Homsy, Joshua Kayiwa, Lisa Nelson, Celine H. Taboy, Stuart Nichol, Pontiano Kaleebu and Jane R. Aceng.

We acknowledge other individuals and organizations that supported and played a role in controlling this outbreak especially, Ministry of Health of Uganda, Kasese District Health Team, US Centers for Disease Control and Prevention (CDC), World Health Organization (WHO), Uganda Red Cross, Médecins Sans Frontières (MSF), African Field Epidemiology Network (AFENET) and Makerere University Kampala.

## Author Contributions

**Conceptualization:** Luke Nyakarahuka, Joel M. Montgomery, Stephen Balinandi, Julius J. Lutwama, John D. Klena, Trevor R. Shoemaker.

**Data curation:** Luke Nyakarahuka, Sophia Mulei, Shannon Whitmer, Kyondo Jackson, Amy Schuh, Jimmy Baluku, Allison Joyce, John D. Klena, Trevor R. Shoemaker.

**Formal analysis:** Luke Nyakarahuka, Sophia Mulei, Shannon Whitmer, Kyondo Jackson, Alex Tumusiime, Amy Schuh, Jimmy Baluku, Allison Joyce, Stephen Balinandi, Trevor R. Shoemaker.

**Funding acquisition:** Jayne B. Tusiime, Joel M. Montgomery, Julius J. Lutwama, John D. Klena, Trevor R. Shoemaker.

**Investigation:** Luke Nyakarahuka, Sophia Mulei, Shannon Whitmer, Kyondo Jackson, Alex Tumusiime, Amy Schuh, Felix Ocom, Jayne B. Tusiime, Stephen Balinandi, Julius J. Lutwama, John D. Klena, Trevor R. Shoemaker.

**Methodology:** Luke Nyakarahuka, Sophia Mulei, Shannon Whitmer, Kyondo Jackson, Alex Tumusiime, Amy Schuh, Jimmy Baluku, Felix Ocom, Jayne B. Tusiime, Stephen Balinandi, John D. Klena, Trevor R. Shoemaker.

**Project administration:** Joel M. Montgomery, Stephen Balinandi, Julius J. Lutwama, John D. Klena, Trevor R. Shoemaker.

**Resources:** Joel M. Montgomery, Julius J. Lutwama, John D. Klena, Trevor R. Shoemaker.

**Software:** Shannon Whitmer, Allison Joyce.

**Supervision:** Jayne B. Tusiime, Joel M. Montgomery, Stephen Balinandi, Julius J. Lutwama, John D. Klena, Trevor R. Shoemaker.

**Validation:** Luke Nyakarahuka, Shannon Whitmer, Joel M. Montgomery, Julius J. Lutwama, John D. Klena, Trevor R. Shoemaker.

**Visualization:** Luke Nyakarahuka, Shannon Whitmer, Jimmy Baluku, Allison Joyce.

**Writing – original draft:** Luke Nyakarahuka, Trevor R. Shoemaker.

**Writing – review & editing:** Luke Nyakarahuka, Sophia Mulei, Shannon Whitmer, Kyondo Jackson, Alex Tumusiime, Amy Schuh, Jimmy Baluku, Allison Joyce, Felix Ocom, Jayne B. Tusiime, Joel M. Montgomery, Stephen Balinandi, Julius J. Lutwama, John D. Klena, Trevor R. Shoemaker.

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
