## [Decision Letter · Decision Letter 0]

13 Dec 2021

Dear Dr Nyakarahuka,

Thank you very much for submitting your manuscript "First Laboratory Confirmation and Sequencing of Zaire ebolavirus in Uganda after introduction of cases from the 10th Ebola Outbreak in the Democratic Republic of the Congo, June 2019." for consideration at PLOS Neglected Tropical Diseases. As with all papers reviewed by the journal, your manuscript was reviewed by members of the editorial board and by several independent reviewers. The reviewers appreciated the attention to an important topic. Based on the reviews, we are likely to accept this manuscript for publication, providing that you modify the manuscript according to the review recommendations. 

We apologize for the long delay with your submission. It has taken an unusually long time to secure reviewers. Please attend to the reviewers' requests for minor revisions. Please note the comment about the abstract; it does not meet journal requirements in length (250-300 words) and purpose.

Sincerely,

Laith Yakob

Associate Editor

Andrea Marzi

Deputy Editor

We apologize for the long delay with your submission. It has taken an unusually long time to secure reviewers. Please attend to the reviewers' requests for minor revisions.

Reviewer's Responses to Questions

**Key Review Criteria Required for Acceptance?**

**Methods**

-Are the objectives of the study clearly articulated with a clear testable hypothesis stated?

-Is the study design appropriate to address the stated objectives?

-Is the population clearly described and appropriate for the hypothesis being tested?

-Is the sample size sufficient to ensure adequate power to address the hypothesis being tested?

-Were correct statistical analysis used to support conclusions?

-Are there concerns about ethical or regulatory requirements being met?

Reviewer #1: (No Response)

Reviewer #2: The objectives are clearly articulated and the study design is appropriate. The population clearly described and appropriate.

Sample size or statistical analyses do=not apply. Ethical or regulatory requirements were met.

**Results**

-Does the analysis presented match the analysis plan?

-Are the results clearly and completely presented?

-Are the figures (Tables, Images) of sufficient quality for clarity?

Reviewer #1: (No Response)

Reviewer #2: The results are well presented.

**Conclusions**

-Are the conclusions supported by the data presented?

-Are the limitations of analysis clearly described?

-Do the authors discuss how these data can be helpful to advance our understanding of the topic under study?

-Is public health relevance addressed?

Reviewer #1: (No Response)

Reviewer #2: The conclusions are well supported by the data. Limitations are clearly described. The usefulness and public health relevance are adequately discussed.

**Editorial and Data Presentation Modifications?**

Reviewer #1: The methods section should precise which Elisa kits were used and the Ebola antigen that were used or targetted in the Elisa assays. The validation criteria (cut-off setting) should be explicite as well if the kit is an in-house one. A reference describing the optimisation of the Elisa assays can also be added.

In addition, the method section should mention the place of other investigations (for example the one which led detection of viruses such as CCHF, RVF...) and present the algorithm. Was the Elisa done only for the PCR positive samples?

Reviewer #2: The abstract should be revised for length andclarity.

**Summary and General Comments**

Reviewer #1: This nice paper is brilliantly written and illustrates how complementary are epidemiological investigations and lab testing or sequencing. It also points out the impact of preparedness program and the role of countries collaboration when faciing a commun threat.

Reviewer #2: Nyakarahuka and coworkers have done a terrific job of detailing laboratory confirmation and sequencing of Ebola virus in Uganda after it was introduced from the Democratic Republic of the Congo. The detail of the epidemiology is striking and important work.

This reviewer has only a few suggestions:

1. Please revise the abstract – it is too long for PLOS NTD and misses the guidance to “summarize the most important results with important numerical results given.” Some of these results are summarized nicely in the last paragraph of the paper. It seems important that “the case detected later in August 2019 (201901815) was from a different lineage from the three related cases in June 2019, demonstrating that the two Uganda EVD outbreaks were independent introductions and not the result of ongoing EBOV transmission in Uganda.”

2. With regard to the two independent introductions, it might be more accurate to revise the title of the paper a bit. 

3. Apologies, but please revise to confirm to the current nomenclature rules with respect to capitalization and use of italics. Also, species of viruses don’t cause disease, members of species do. The title should probably just say Ebola virus, not the species name.

PLOS authors have the option to publish the peer review history of their article (what does this mean?). If published, this will include your full peer review and any attached files.

Reviewer #1: No

Reviewer #2: No

Figure Files:

Data Requirements:

Reproducibility:

References

---

## [Editor Report · Decision Letter 1]

26 Jan 2022

Dear Dr Nyakarahuka,

We are pleased to inform you that your manuscript 'First Laboratory Confirmation and Sequencing of Zaire ebolavirus in Uganda following two independent introductions of cases from the 10th Ebola Outbreak in the Democratic Republic of the Congo, June 2019' has been provisionally accepted for publication in PLOS Neglected Tropical Diseases.

Best regards,

Laith Yakob

Associate Editor

Andrea Marzi

Deputy Editor

---

## [Editor Report · Acceptance letter]

15 Feb 2022

Dear Dr Nyakarahuka,

We are delighted to inform you that your manuscript, " First Laboratory Confirmation and Sequencing of Zaire ebolavirus in Uganda following two independent introductions of cases from the 10th Ebola Outbreak in the Democratic Republic of the Congo, June 2019," has been formally accepted for publication in PLOS Neglected Tropical Diseases.

Best regards,

Shaden Kamhawi

co-Editor-in-Chief

Paul Brindley

co-Editor-in-Chief
